# Adaptive Feeding Roller with An Integrated Cutting System for Automated Fiber Placement (AFP)

## Mohammad Bahar *[ID] and Michael Sinapius[ID]

Institut for Adaptronics and Function Integration, Technical University of Braunschweig, Langer Kamp 6, 38106 Braunschweig, Germany; m.sinapius@tu-braunschweig.de

* Correspondence: m.bahar@tu-braunschweig.de

**Abstract:** Automated Fiber Placement (AFP) has been used for more than 30 years as a manufacturing process for large components such as aircraft fuselage or nose made of composite materials. As the literature research showed, 85% of efficient AFP head process time is used for the cutting process, accelerating, decelerating and necessary directional changes of the AFP head. The AFP head consists of various modules that synchronize the complete process. In recent years, scientists tried to extend the advantages of this technique and optimize the process. The aim of this article is to show the integration of the cutting unit and the feeding unit in a compact and simple system. Therefore the cutting edge of the tape and the productivity can be increased significantly. The feeding system pulls the tape from the material spool to under the compaction roller. Obviously, the tape should also cut to a predefined length. According to the state of the art, the cutting unit works when the feeding unit is in the standing position. The layup process has therefore a short break, which is no longer required with the new approach. The cutting process is realized without interruption, therefore the lost time during cutting can be saved. The integrated unit is synchronized with the AFP process to apply the active or passive feeding process.

**Keywords:** Automated Fiber Placement; adaptive feeding roller; AFP cutting system; AFP feeding system; Automated Tape Laying; 3D print feeding system

---

## 1. Introduction

Fiber-reinforced plastics (FRP) are increasingly used in many industrial applications. FRP is suitable for many applications in the aerospace and automotive industries. Especially in civil aviation different placement technologies are being developed to achieve lightweight, more ecological and efficient airplanes. For this reason, the new technologies, such as Automated Fiber Placement (AFP) and Automated Tape Laying (ATL) is being improved very intensively [1]. The AFP process (Figure 1) enables the quality of the laminate to increase significantly while reducing costs and production time. The modular AFP head consists of a material supply unit, feeding unit, cutting mechanism, guiding unit, heating system, and compaction unit [2]. Many researchers aim to increase the laying speed concerning high laminate quality. For the production of complex and small parts, tapes have to be cut more often and the machine must accelerate continuously in different axes. Researches in 16th Machining Innovations Conference for Aerospace Industry in 2016 published that 46% of the time is spent by placing slit tapes. The remaining 54% is used for plies inspecting and defects correction, setting up and cleaning.Additionally they determined that the tape cutting process is the most time-consuming stage. 85% of the placing time is spent for stopping the head, cutting the tape and again accelerating and decelerating. That leads to the conclusion that the cutting unit in AFP requires further improvement [1].

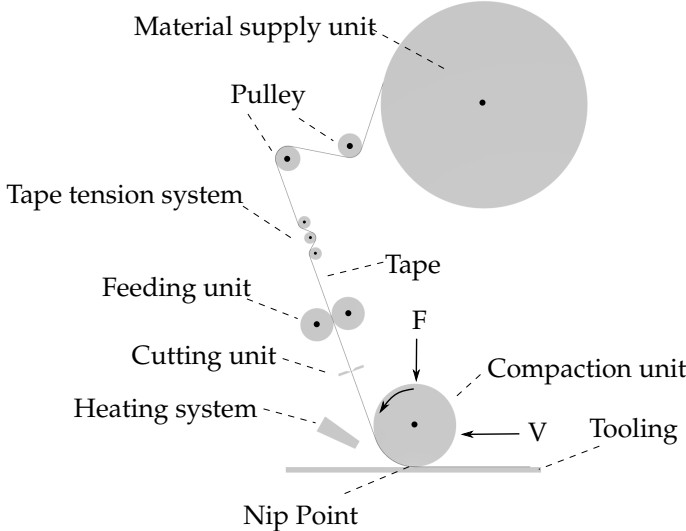

**Figure 1.** Automated Fiber Placement (AFP) process.

To improve productivity in AFP applications, a variety of different cutting and feeding techniques can be used to manufacture CFRP composite structures. The six typical cutting methods are shown in Figure 2. All six different techniques have their advantages and disadvantages. As it is shown in the Figure 2a, the feeding and cutting units can be combined such as one blade cutting unit is fixed below the feeding unit. The cutting unit should be placed as close as possible to the nip-point (def.:Compaction point between the incoming tape and the deposited tape) if this is required to achieve a short tape placement. It is mainly used for the production of small and complex CFRP composite structures. The most common variant is when the carbon fiber prepreg is cut with a cutting unit before passing through the feeding unit, as shown in Figure 2b. It reduces the distance between the feeding unit and the compaction unit to a minimum and keeps the CFRP prepreg tensioned for the maximum period of time. It is possible to use a two blades system and the cutting unit. This unit can be mounted below or above the feeding unit. However, this method is not the most effective method regarding complex design. These combinations are shown in Figure 2c–d. The Holmatec Maschinenbau GmbH [3] Company tested different techniques of cutting carbon fiber and they found out that precise cuts and low tool wear can be reached by using a two blades cutting unit. As it is shown in Figure 2e, one blade cutting unit can be integrated into the feeding unit roller. It allows to increase free space in the AFP head, minimize the distance between the feeding unit and compaction unit and keeps the CFRP prepreg tensioned for the maximum time period. However, using this method the placement paths length is not minimized, because the cutting unit is not located as close to the nip-point as possible. The laser cutting system can be used instead of the blade cutting system, as shown in Figure 2f. This method achieves a high-quality cut and can be installed above and below the feeding unit. This process is very energy-intensive, the laser radiation is reflected, absorbed and transmitted by the interaction with the material. Freitag et al. used ray tracing method to determine that absorption during the laser cutting of carbon fiber is about 75% at a wavelength of 1064 nm. Fuchs et al. have shown that depending on the fiber angle, in a range between 9% to 40% of laser power was transmitted and lost during laser cutting of carbon fiber [3–5].

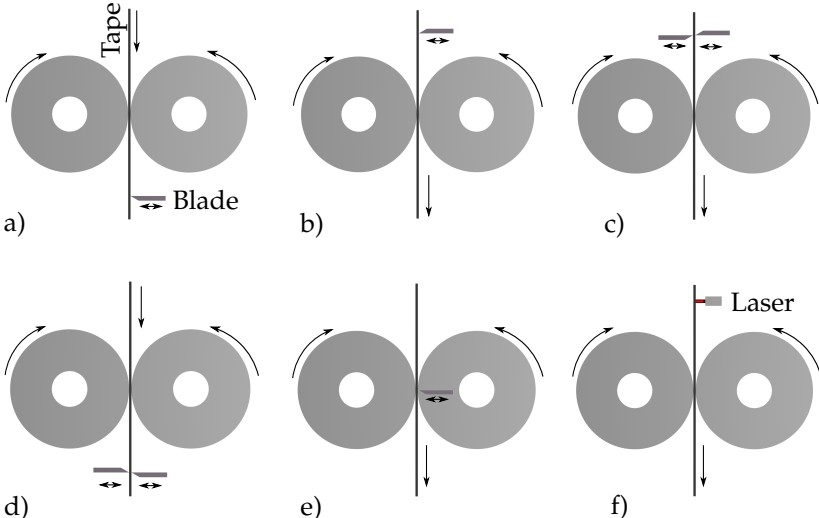

**Figure 2.** Configuration of different cutting methods.

*Current Cutting Systems*

To achieve the required productivity and quality in the various AFP applications, there is a wide range of different cutting and feed techniques. Figure 3a shows a MTorres cutting system in which several tapes can be cut in parallel and independently of each other. The cutting is realized by a Guillotine configuration. The tape (3) is transferred to the compaction roll (18) by the AFP head and laid down there. A swiveling blade (19) is attached in the middle, which cuts the tape by shearing it off on a one-sided support. The blade is moved by a horizontal sliding movement in a guide rail. In this way, several blades can be applied side by side and controlled independently of each other [6]. Start-cut-restart System from MTorres AFP head with a rotary cutting system concept is shown in Figure 3b in two views. The head cutting unit (5) with a blade (14) integrated inside the rotating roller (15) allows the cutting of the multiple numbers of fibers (1) when it is required to break their application on the placement surface. The rotating roller (15) and each separate blade (14) are connected to the driving ring (4.1) and synchronized actuated through the coupling system (16). With a complete rotation of the driving ring (4.1), the start-cut-restart process can be done. It is only important to arrange the recesses (12) of the driving ring (4.1) and the cutting unit (5) with a blade (14) in a required angular position [7]. Denkena et al. developed an AFP system for CFRP tapes placement. In this AFP-Head, the researchers use an anvil-blade cutting unit system, which is shown in Figure 3c. To increase the blades lifetime, the cutting blade (anvil) is made from self-healing material. To achieve different geometries, each tow is cut by a separate blade using pneumatic actuators. Fast and short cylinder movement is controlled by switching valves and cutting can be done in a few milliseconds. Furthermore, the heating unit with four infrared heaters is integrated into a cutting unit to heat up the surface of placed CFRP tape [1].

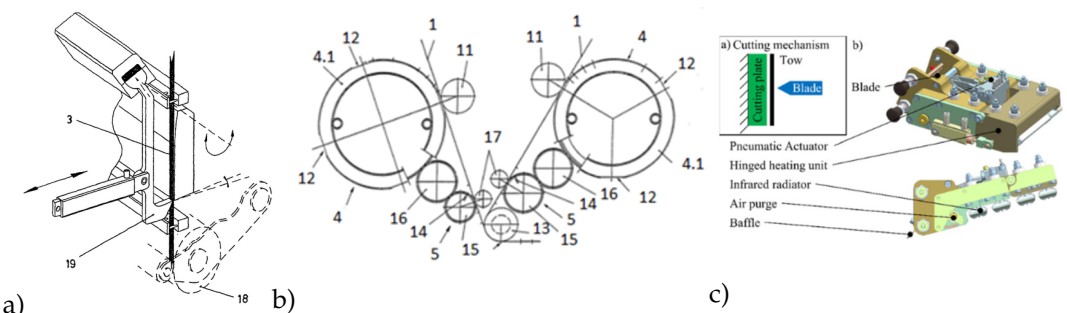

**Figure 3.** Schematics of the different cutting systems [1,6,7].

In some concepts, each module is mounted separately, in turn these designs require an additional space in the AFP head. Moreover, an additional module (tape buffer) is integrated in an AFP head in order to reduce the tape feeding delay after cutting [8]. The complex design with large construction size leads to limitations of the AFP process. To increase efficiency in production time, A.E. Modin et al. published a new design of the cutting mechanism patent for AFP machines in 2016 [9]. These cutters have multiple numbers of double-sided blades, which allows them to cut fiber tows in different directions. The blades are moved back and forth to cut in both instead of a single direction. In that case the number of cuts and the time of tool wearing is doubled. It also reduces the time for restarting the blade to a ready position after the first cut. The blade includes a lower edge, which is oriented to cut when the blade is driven down, and an upper edge, which is oriented to cut when the blade is driven upward.

The cutting unit is not the only essential element, but also the blade alignment and tape orientation are decisive for a high-quality cutting process. Many experiments served to determine the defect dependency to the orthogonal cutting in regards to fiber orientation [10,11]. The influence of the cutting principles through the cutting of full edge and crossing edges is an important factor. According to microscopic analysis, published by Henneberg [12] it was found that better tip quality of CFRP tape with a 90° fibers orientation can be reached by using full edge cutting technique.

## 2. Materials and Methods

In this study, the placement speed is increased by integrating the feeding unit and the cutting unit. The enhanced speed of the tape laying process is not the only advantage in this method. An adaptive feed roller allows the tape tension to be simply adjusted to the requirements during the laying process. On the one hand, the free end of the tape can be transported to the nip point by using the stiffness of the tape, on the other hand, the placement length can be minimized. An additional field of application for an adaptive feed roller with the integrated cutting system is the automation of 3D printing with continuous carbon fiber. There is currently no sufficiently available solution for these types of 3D printers. Fiber material and filament are transported by different systems and joined together in the printing head. In contrast to conventional printing filaments, the carbon fiber filament has to be cut after printing [13]. A combined feeding and cutting unit can also be integrated for 3D printing with continuous carbon fiber.

An adaptive feed roller is required to combine the feeding and cutting functions in one system. The feeding roller has the function to switch between active and passive feeding. Figure 4 shows the active feeding system and the passive feeding system in 3 steps.

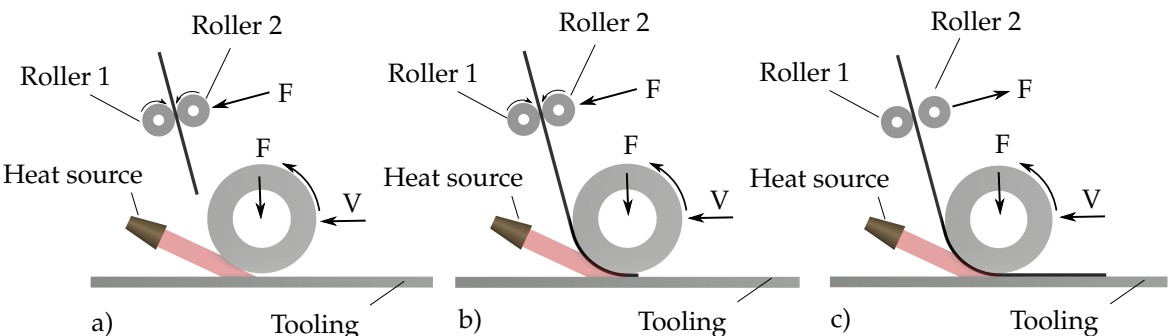

**Figure 4.** Functional principle Active and Passive Feeding System in AFP.

In the first step shown in Figure 4a, the tape is transported to the compaction roller using both rollers. The driven roller 1 is fixed and the roller 2 is movable perpendicular to the tape. Roll 2 presses the tape and rotates together with roller 1 and the tape is transported through an active feeding system from material spool to compaction roller. In step (b), the tape is transported with

the active feeding system under the compaction roller. The tape is activated by a heat source and can be applied to the tool. In steps (a) and (b), the tape tension between material spool and feeding unit is measured by a tape tension sensor and can be adjusted by a hysteresis brake, installed in the material spool. In these two steps, the tape tension can be less than the tape tension in the placement process. These steps aim to transport the tape to the nip-point. The tape can be placed by rotating and moving the compaction roller. The adhesive between the tape and the tool is then increased and the tape does not require active feeding. This allows the tape to be transported through material spools as the AFP head moves. In this step (c) roller 2 has no contact with the tape and roller 1 is inactive. In the case of passive feeding, the tape tension is measured again by the tape tension sensor and adjusted with the hysteresis brake. An active and passive feeding system allows the tape tension to be kept constant as an important parameter in the AFP process.

Figure 5a shows the schematic view of a feeding roller with the integrated blade. The two rollers are fixed on two shafts and roller 1 can be rotated by a stepper motor. The stepper motor defines the feeding speed for the active feed. By installing the gear unit on roller 1, the rotation can be transmitted via the gear unit on roller 2. These gears guarantee that the blade position and the blade pocket in roller 1 are always synchronized with each other. The servo motor can adjust the position of roller 2 as an active or passive feeding system. The tape length and the tape speed are continuously measured by an integrated optical sensor (Figure 6b), which is installed behind the feeding system. When the blade position is in Pos. 1, the tape is moved forward by an active feeding. At the beginning of the passive feeding, the two rollers have no contact with each other until shortly before the tape is cut. The speed of roller 1 and the placement speed are synchronized by the control system. The passive feeding is switched to active feeding shortly before the tape is cut. At the same time, the blade moves from Pos.1 to Pos.2 and the cutting process is complete (Figure 5b). The blade influences the cutting edge of the tape. The investigation published in [12] shows that the asymmetric blade has a better cutting quality compared to the symmetric blade and how to align the blade on the tape.

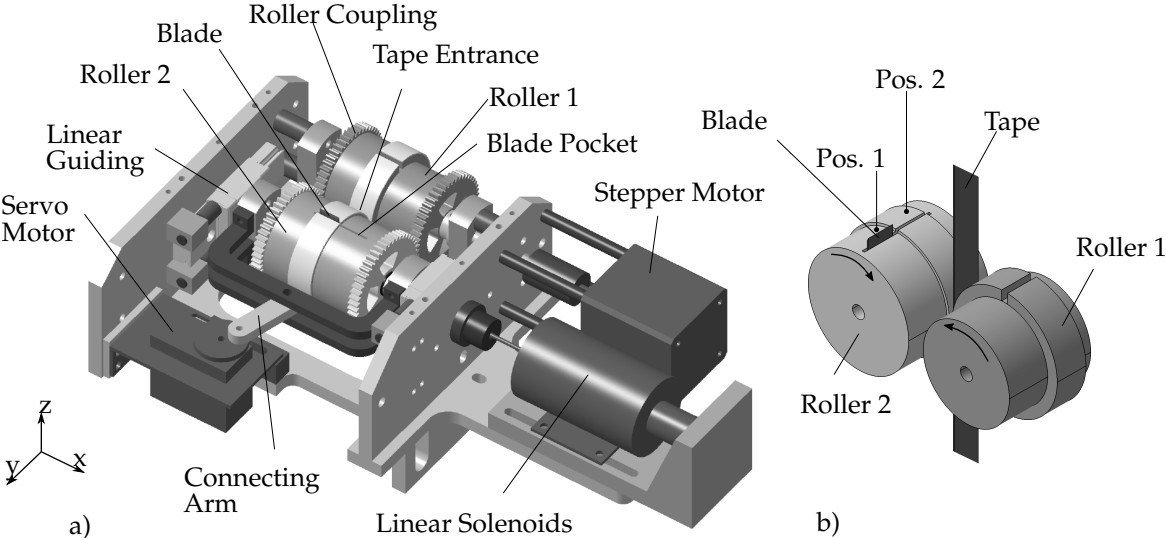

**Figure 5.** Schematic design of an integrated feeding and cutting unit for an AFP head.

In Figure 6a, the roller 2 is illustrated in detail. The hollow shaft is supported by two pillow blocks and the roller 2 and the roller coupling is installed on the hollow shaft. The movable shaft, attached with a linear solenoid, allows the blade holder to move in the cutting stage.

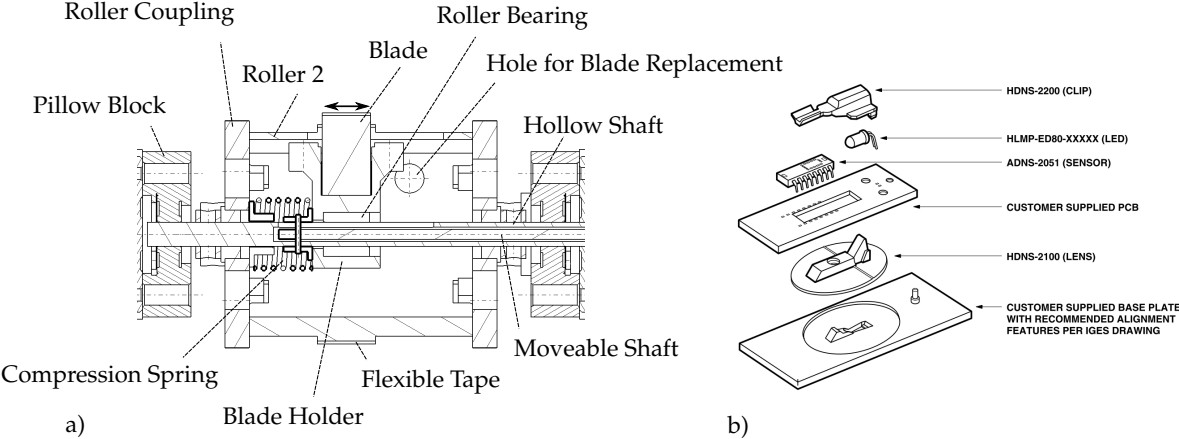

**Figure 6.** Schematic principle of the adaptive feeding roller and the optical sensor [14].

When the cutting process is completed, the blade holder retracts via a compression spring installed on the other side of the blade holder. Furthermore, the blade can easily be changed through a hole integrated on roller 2. To increase the frictional force between the roll and the protection of the tape surface during feeding, a flexible tape consisting of PA6 and textiles is applied to the rollers.

The drive torque of the feeding unit by using feeding between two rollers technique can be calculated via Equation (1).

$$M_D = M_L.K = F_T.K.r_D \tag{1}$$

Here: $M_L$—load torque, $Nmm$ ; $F_T$—required tape tensile force, N; K—safety factor; $r_D$—radius of the drive roller, mm. According to the CF-PA6 CFR unidirectional tape specifications, the minimum tensile force is recommended to be $F_T$ = 2.26 N. To generate a small shaft torque, the drive roller diameter must be kept as small as possible. At the same time, the width of the tape should be kept in mind for proper tape insertion for choosing the right length roller. The minimum radius of the drive and addition roller can be assumed to be $r_{Dmin}$ = 15 mm. The maximum radius of the drive roller should not exceed 30 mm, considering the requirements for the dimension $r_{Dmax}$ = 30 mm. With the safety factor K = 2, Equation (1) gives the minimum drive torque $M_{Dmin}$ and the maximum drive torque $M_{Dmax}$:

$$M_{Dmin} = F_T.K.r_{Dmin} = 2.26 * 2 * 15 = 67.8 Nmm \tag{2}$$

$$M_{Dmin} = F_T.K.r_{Dmin} = 2.26 * 2 * 30 = 135.6 Nmm \tag{3}$$

The tape feeding is detected by the ADNS-2051 [14] optical sensor. Usually this sensor is used to detect the position of optical mouses and in this implementation can detect the feeding, the tape speed and the position of the tape end. Therefore the sensor is installed together with a light source on a base frame and integrated into the tape guide behind the feeding unit. The sensor provides high-speed motion detection with a pixel resolution and a high-speed image frequency of 2000 to 6400 Hz refresh rate, which ensures a precise determination of tape position and speed. The sensor has an SPI interface and can be combined with most microprocessors. Figure 6b shows an exploded view of the optical sensor with all required components. This concept is developed only for one tape spool.

The general concept of the working principle for the adaptive feed roller with an integrated cutting system is based on the algorithm, which is shown in Figure 7. Regarding the automatically generated placement path, the number of required layers, and the length of each layer is calculated. The first layer placement starts by turning on active feeding. CFRP tape starts moving and synchronously measurement of placed layer length is continuously assured by the optical flow sensor. When the tape tip reaches the nip-point and the possibility of losing the tape is avoided, the tape tension measurement between active feeding and passive feeding units starts.

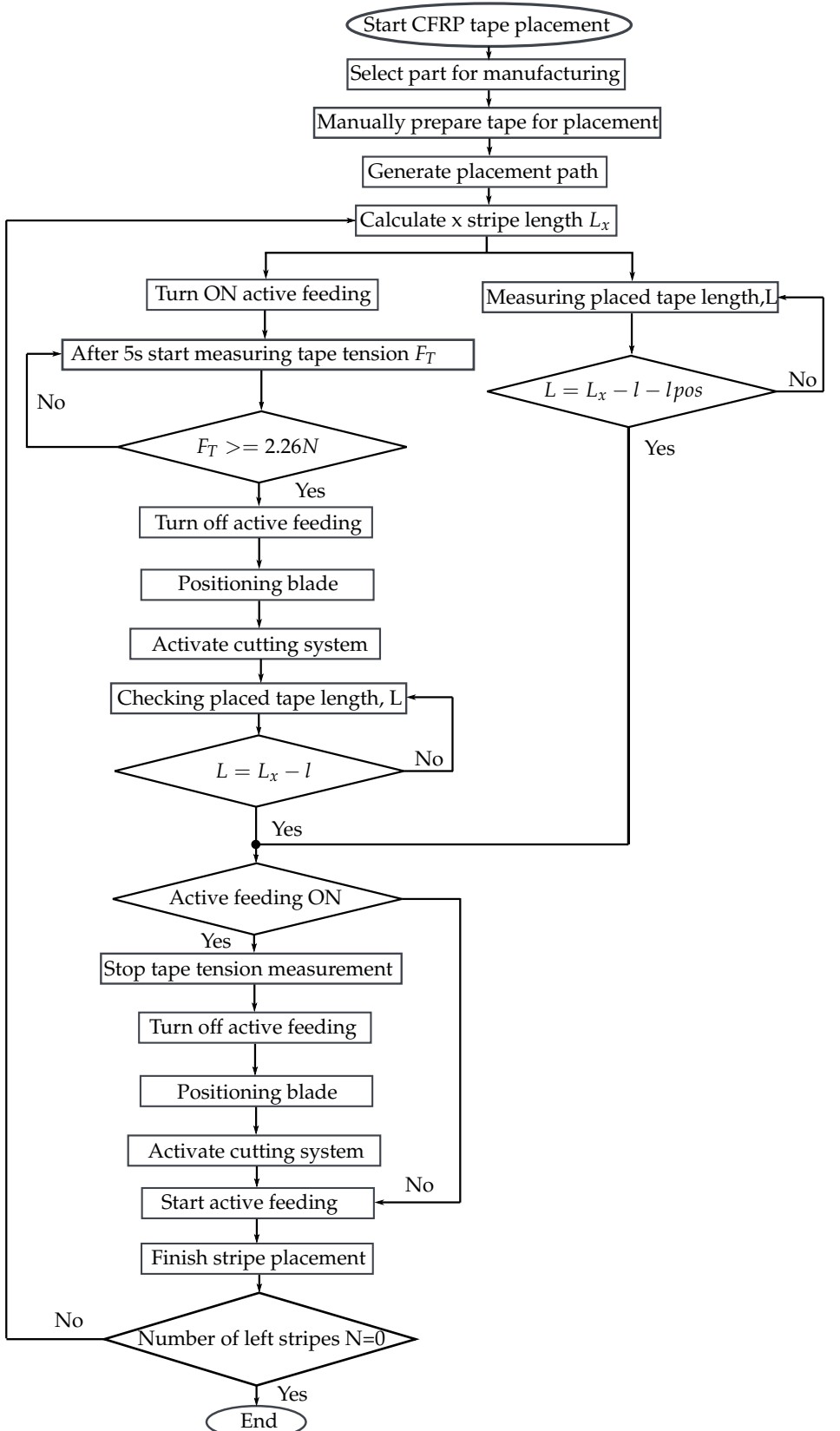

**Figure 7.** Functional algorithm for the adaptive feed roller with an integrated cutting system for automated fiber placement (AFP).

For this, at least 5 s interval is required and mentioned in the working algorithm. When the tape tension reaches, active feeding is turned off and the placed tape length is checked until it reaches the required length. Then, the blade is positioned for the cutting process. The tape tension at that point is controlled by using the material supply unit brake element. The passive feeding lasts until the measured placed tape length is equal to $L = L_x - l$, where l is the constant known distance between the positioned blade and cutting point. To activate the cutting unit, an active feeding roller starts rotating, the tape is cut and is placed on the surface. During the composite structures manufacturing process, it is unavoidable to manufacture small parts or complex surfaces, and the short layer placement is required. For that second loop is developed, where required placed tape length is reached before the tape tension reaches $F_T \geq 2.26$ N. In that case, active feeding lasts until the measured placed tape length is equal to $L = L_x - l - lpos$ where l is the constant known distance between the positioned blade and cutting point and $lpos$ is the length of the moving tape, while the blade is being positioned. To activate the cutting unit, an active feeding roller starts rotating, the tape is cut and is placed on the surface. However, during the AFP process, tape tension between the feeding unit and the material supply unit must reach $F_T \geq 2.26$ N during the 5 s time interval, and the previously discussed loop is mostly used just as a safety factor. The loop lasts until the number of calculated required lays is equal to N = 0 and the AFP process for the manufacturing part is over.

## 3. Results and Discussion

After validation and evaluation of the manufacturing and programming of the control system, the new concept is realized and investigated experimentally. In this step, 6.3 mm wide CFRP-PA6 and CFRP-PEEK unidirectional carbon fiber tapes are used to evaluate the cut quality in relation to the feeding speed. Different speeds between $v_{min} = 4\frac{m}{min}$ and $v_{max} = 12\frac{m}{min}$ are investigated according to the different feeding speed requirements. For feeding rollers with $R = 21$ mm, the feed motor speed is set in the range from 30 rpm to 90 rpm and several cuts are implemented for each feeding speed in order to be able to compare the cutting quality with each other, depending on the feeding speed. Regarding the material properties, the tapes can be cut with 1 Nm torque. All samples are cut successively with one blade.

A flexible adhesive PA6-tape is applied to both rolls to increase friction between two rollers and tape without damaging the tape. The tape can be transported by active feeding and then cut to the required length. Through several experiments, the tape can be moved forward at different speeds up to $v_{max} = 60\frac{m}{min}$ without any further impairment. For the first step, the tape is inserted manually between the two rollers. The tape is then automatically moved forward by the feeding system. After many cuts, the blade is no longer sharp in order to complete the cutting process successfully. With this concept, the blade can easily be replaced by a screw integrated into the roller unit. Figure 8 shows a prototype model for the adaptive feeding roller with an integrated cutting system for AFP.

Experimental results for feeding speeds of $4\frac{m}{min}$, $8\frac{m}{min}$ and $12\frac{m}{min}$ investigated by microscopic analysis are shown in Figure 9. Considering the results achieved with the adaptive feeding roller, the feeding speed influences the length of the fiber cracks. The driven roller and blade pressure is the main cutting force and defines the fracture morphology of the CFRP tape. One of the most important parameters of the AFP process is the tape tension. Due to the pressing force between the two rollers, the tape tension between the material spool and the feeding system as well as after the cutting process is successfully tested.

Figure 9 shows that very high quality, essentially constant cutting edge, can be achieved at different speeds. The fiber crack length after cutting is mostly between 0.05–0.1 mm, which guarantees a successful cutting process. The breaking elongation of PEEK materials is one-third of PA6 [15], therefore the cut edge quality of CFRP-PEEK is significantly better than that of CFRP-PA6 materials. The feeding speed can be increased up to $60\frac{m}{min}$, although it is a challenge for an in-situ consolidation. However, a different stepper motor must be used for this purpose, as the speed of the current stepper motor is not sufficient for high speed applications. In addition, an optical sensor such as an optical

mouse sensor can be integrated after the feeding unit to measure the tape length with an accuracy of less than 0.05 mm.

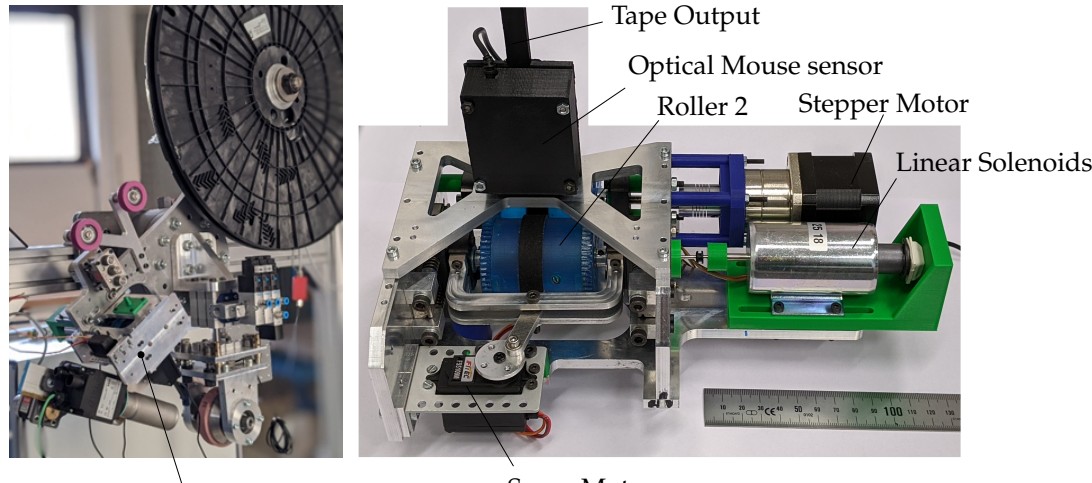

**Figure 8.** Prototype model for an Adaptive Feeding Roller with an Integrated Cutting System for AFP.

| Feeding speed $4\frac{m}{min}$ CFRP-PA6 | Feeding speed $8\frac{m}{min}$ CFRP-PA6 | Feeding speed $12\frac{m}{min}$ CFRP-PA6 | Feeding speed $12\frac{m}{min}$ CFRP-PEEK |
|---|---|---|---|
| 6.35 mm | | | 6.35 mm |
| | | | |
| | | | |
| | | | |
| | | | |

**Figure 9.** Microscopic view of the cutting edge in relation to the feed speed (cut on top).

## 4. Conclusions

This research is focused on improving the AFP process by developing a new design concept for the feeding and cutting unit that improves the process efficiency regarding the process time. The literature research shows that 85% of the efficient AFP process time is used for the cutting process, acceleration and deceleration of the AFP head [1]. To increase AFP productivity, a cutting unit has been integrated into a feeding unit. With an active and passive feeding system, the tape can be cut without interruption. This method allows the tape tension to be adjusted during the placement process. An important factor in the AFP process is the cutting quality of the tape. The cutting quality is tested with different tape speeds up to $12\frac{m}{min}$ for CFRP-PA6 and CFRP-PEEK materials. The fiber crack length after cutting is between 0.05–0.1 mm, which guarantees a successful cutting process. During the cutting process, the tape surface remains undamaged. The integrated system is simple to install and the tape can be cut to the desired length. this concept can also be extended and used for placement on multi two fiber.

**Author Contributions:** M.B. conceived, designed and performed the experiments. M.B. and M.S. commonly analyzed the data and wrote the paper. All authors have read and agreed to the published version of the manuscript.

**Funding:** The authors acknowledge the funding of the study by "Deutsche Forschungsgemeinschaft. We acknowledge support by the German Research Foundation and the Open Access Publication Funds of the Technische Universität Braunschweig.

**Conflicts of Interest:** The authors declare no conflict of interest.

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
