# Peer review of "Adaptive Feeding Roller with An Integrated Cutting System for Automated Fiber Placement (AFP)"

_jcs, doi:10.3390/jcs4030092_

Round 1

Reviewer 1 Report

The abstract need to be re written in proper english. Besides that this is just a technical disclosure. Article whre the authors have tried to show the efficacy of the unique fiber cutting system.

With the new system why was there now work done at actual afp production speeds. Also at those speed is there a possibility that the feed tape might buckle or fold up due to the lack of tape stiffness caused by the heat source right before the tape touches tolling and the nip of the compaction roller.

What is missing is any sort of detailed study of the laminate integrity produced by this system or any indepth SEM work on Fiber integrity. Please include more data to support the efficacy of the system

Author Response

Hello dear reviewer,
first of all, thank you for your suggestion and comments

I have listed the changes in the text and for some of the suggestions I have an explanation:

This method works with an active and a passive system, as I explain in the research. the heat source can be active when the tape is already under the compaction roller.figure 4

The speed of the depositing process is dependent on the AFP head and heat source. With this method, it is possible to increase the depositing speed.

I'm working on the AFP head, and the AFP head isn't ready for placement.

With kind regards

Reviewer 2 Report

In this paper, the integration of the cutting unit and the feeding unit in a compact and simple system for Automated Fiber Placement (AFP) is demonstrated. The paper falls well within the scope of Journal of Composite Science. The subject area has already been widely investigated and reported (patents and other articles). The Authors have not been able to bring out the novel aspect of the paper. There are papers in the exact same area. The contribution to knowledge base is little, and as such the paper should be declined.

Author Response

Hello dear reviewer,

Automated Fiber Placement is a well-known procedure, but there are very few publications on feeding or cutting system.

and only 3 patent on the feed and cutting system without break for AFP process from Companies. In this work, I have shown how the depositing process is possible without a break through a simple system. The different feed speeds were investigated and the quality of the edge prepregs after cutting was shown.

With kind regards

Reviewer 3 Report

In this manuscript, the authors design an adaptive feeding roller with an integrated cutting system for AFP. The experiment results showed that the integrated system exhibits a successful cutting process and the fiber crack length after cutting is between 0.05-0.1 mm. This paper may be accepted in the Journal of Composites Science with minor revision. The following points should be clarified before accepted:

1) Please label all the parts of the prototype model in Figure 7 (for example roller 1, roller 2, blade, etc.).

2) Please provide the microscopic view of the original CFRP-PA6 and CFRP-PEEK without cutting to compare the tape surface before and after the cutting process.

3) Please provide the cross-section view of the cutting edge in relation to the feed speed. 

4) Please keep the format of the citing figure names consistent. Both “fig.” and “Fig.” appear in the main manuscript.

5) In Page 7 Line 190, the authors assert that:

“The feeding speed is successfully tested up to 60 m/190 min …” Please provide any evidence to support this assertion.

Author Response

Hello dear reviewer,
first of all, thank you for your suggestion and comments

I have listed the changes in the text and for some of the suggestions I have an explanation:

1-done.

2- The Prepreg is made of continuous carbon fiber and is not cut. The original cutting is done manually by the manufacturer using a cutting machine.

3- Prepreg has a thickness of 0.17 mm and the normal view of Prepreg is not relevant for this study.

4-done

5-done

Round 2

Reviewer 1 Report

though this is not the best explanation I was looking for and since this is developmental work, I look froward to more work building on these initial finding 

Author Response

Thank you for your feedback.
i have added 2 pages about the process details and control of the system.

Reviewer 2 Report

The issue of novelty and contribution to existing knowledge remains as it is. The paper does not present anything novel, here. In the Introduction, nowhere a reader finds the research gaps or the mentioning of Authors' strategy to cope with it.
The importance of the paper is low, and as such it should be declined.

Author Response

Thank you for your feedback.
I have added 2 pages about the process details and control of the system.

the integration between the feeding system and cutting system is an innovative method that is not presented in any paper.

Reviewer 3 Report

The authors did not answer all of my questions.

(1) No additional evidence (e.g. Microscopic view of the cutting edge in relation to the feed speed 60 m/min) was provided in the main manuscript.

(2) Both "fig" and "Fig" appeared in the main manuscript. 

Author Response

Thank you for your feedback.
I have added 2 pages about the Details of the system and the control.

the microscopic image normal to tape is not relevant and has no advantages, because after discarding this area will be cut and has no influence on the part